# Validation of the Spanish Version of the Fear of COVID-19 Scale (FCV-19S) in Long-Term Care Settings

**DOI:** 10.3390/ijerph192316183

**Published:** 2022-12-03

**Authors:** Pilar Cárdenas Soriano, Carmen Rodriguez-Blazquez, Maria João Forjaz, Alba Ayala, Fermina Rojo-Perez, Gloria Fernandez-Mayoralas, Maria-Angeles Molina-Martinez, Carmen Perez de Arenaza Escribano, Vicente Rodriguez-Rodriguez

**Affiliations:** 1Department of Preventive Medicine, University Hospital of Albacete, ES-02006 Albacete, Spain; 2National Centre of Epidemiology and Network Centre for Biomedical Research in Neurodegenerative Diseases (CIBERNED), Carlos III Institute of Health, ES-28029 Madrid, Spain; 3National Centre of Epidemiology and Health Service Research Network on Chronic Diseases (REDISSEC) and Research Network on Chronicity, Primary Care and Health Promotion (RICAPPS), Carlos III Institute of Health, ES-28029 Madrid, Spain; 4Department of Statistics, University Carlos III of Madrid, and Health Service Research Network on Chronic Diseases (REDISSEC), Carlos III Institute of Health, ES-28029 Madrid, Spain; 5Grupo de Investigacion Sobre Envejecimiento (GIE), IEGD, CSIC, ES-28037 Madrid, Spain; 6Faculty of Psychology, National University of Distance Education (UNED), ES-28037 Madrid, Spain

**Keywords:** fear, COVID-19, long-term care, older people, psychometric properties, Rasch analysis, classical test theory, Fear of COVID-19 Scale (FCV-19S)

## Abstract

Fear of coronavirus disease 2019 (COVID-19) is one of the main psychological impacts of the actual pandemic, especially among the population groups with higher mortality rates. The Fear of COVID-19 Scale (FCV-19S) has been used in different scenarios to assess fear associated with COVID-19, but this has not been done frequently in people living in long-term care (LTC) settings. The present study is aimed at measuring the psychometric properties of the Spanish version of the FCV-19S in residents in LTC settings, following both the classical test theory (CTT) and Rasch model frameworks. The participants (n = 447), aged 60 years or older, were asked to complete the FCV-19S and to report, among other issues, their levels of depression, resilience, emotional wellbeing and health-related quality of life with validated scales. The mean FCV-19S score was 18.36 (SD 8.28, range 7–35), with higher scores for women, participants with lower education (primary or less) and higher adherence to preventive measures (all, *p* < 0.05). The Cronbach’s alpha for the FCV-19S was 0.94. After eliminating two items due to a lack of fit, the FCV-19S showed a good fit to the Rasch model (χ^2^ (20) = 30.24, *p* = 0.019, PSI = 0.87), with unidimensionality (binomial 95% CI 0.001 to 0.045) and item local independency. Question 5 showed differential item functioning by sex. The present study shows that the FCV-19S has satisfactory reliability and validity, which supports its use to effectively measure fear in older people living in LTC settings. This tool could help identify risk groups that may need specific health education and effective communication strategies to lower fear levels. This might have a beneficial impact on adherence to preventive measures.

## 1. Introduction

The pandemic caused by the SARS-CoV-2 virus has had severe repercussions in older adults as they are one of the most vulnerable groups hit by COVID-19 [1]. Not only has it caused, as of September 2022, more than 600 million confirmed cases and around 6.5 million deaths worldwide [2], but it is also responsible for multiple disabilities and infection sequelae. COVID-19 has had a remarkable impact on mental health, and it has become a priority for clinicians, patients and their families and caregivers.

In Spain, almost 20% of the population is aged 65 years or older. The dependency ratio was 29.20% in 2018, and it will likely increase in the coming decades [3]. In this context, long-term care (LTC) settings have been one of the most affected environments, due to the large amount of dependent older adults with co-morbidities that share the same space, which can facilitate the spread of infection [4]. It is especially important in older people living in long-term care settings where the mortality due to SARS-CoV-2 was 10.6% by May 2021 [5].

Infection fatality rates for COVID-19 [6] and severe disease [7] increase exponentially with age, which translates into a higher risk of suffering COVID-19 or its sequelae for older people [8]. As a vulnerable group, they might have a tendency to be more fearful of the disease [9]. Furthermore, the impact that fear, social distancing measures and other aspects of the pandemic might have had on residents’ mental health is still uncertain.

There are few tools to assess psychological aspects in the context of new health threats, such as fear, resilience or coping strategies, which are linked to a population’s behavior by different mechanisms. Furthermore, it is widely accepted that the way we behave as a population can have an effect on an increase or decrease in viral transmission [10]. Since the beginning of the pandemic, several efforts have been made to understand people’s perceptions and cultural insights, as these are a key factor in implementing well-received policies that can reduce the impact of COVID-19 [11].

Acceptance of and compliance with the preventive measures against COVID-19 have been studied in the general Spanish population [12], as well as the demotivation to follow such measures (pandemic fatigue) [13]. Therefore, there is an urge to obtain, understand and evaluate information about behavioral insights and psychological states, specifically fear, in older people, to establish recommendations and interventions in the present pandemic context or in future scenarios.

Fear of COVID-19 can be quantitatively measured using the Fear of COVID-19 Scale (FCV-19S) [14], a scale specifically developed by Ahorsu et al. in Farsi for the pandemic, and which has been recently validated in Spanish [15]. Since its publication, the FCV-19S has been applied in different countries and contexts [16,17]. Nevertheless, the scale has not been frequently used to assess fear among older people in LTC settings [18]. This segment of the population is known to have suffered more strict preventive measures than any other [19]. Getting to know their level of fear would help to make an initial diagnosis of the emotional impact that COVID-19 has had on them.

In this study, we aimed to measure the psychometric properties of the Spanish version of the FCV-19S applied in residents in LTC setting of the Community of Madrid (Spain) by using both the classical test theory (CTT) and Rasch model approaches.

## 2. Materials and Methods

### 2.1. Study Design and Setting

A cross-sectional regional study in a sample of older adults living in LTC settings participating in the project “Nursing homes and COVID-19. Environments of the older people as protectors in health emergency situations (COVID-19)” took place from June to October, 2021. Further information about the project development, descriptive data and sample selection criteria can be found in the publication by Rodriguez-Rodriguez et al. [20].

### 2.2. Participants

Four hundred and forty-seven people aged 60 years or older living in an LTC setting at the Community of Madrid comprised the sample. Participants were excluded if they were aged less than 60 years, suffered from cognitive impairment (since the use of self-reported scales implies having cognitive abilities sufficient to understand and respond to the questions) or refused to sign an informed consent form. All participants signed the informed consent form by themselves, as they were cognitively intact.

### 2.3. Measures

#### 2.3.1. Sociodemographic Features

Information on sex (male or female), age, marital status (married or living as a couple, separated or divorced, single, or widow/er) and level of education (cannot read or write, can read or write but no primary education, primary education, secondary or vocational education, or university education) were collected using the questionnaire. Clinical information, such as diagnostics and prescribed medication, were compiled from their medical history to characterize the sample.

#### 2.3.2. FCV-19S

The main measure was the fear of COVID-19 measured with the FCV-19S scale, a short seven-item self-reported scale that was developed in 2020. Its items are based on the review and assessment of 30 other “fear scales”, and they are answered with a five-point Likert-type scale from 1 (strongly disagree) to 5 (strongly agree). A higher score on the FCV-19S indicates a higher level of fear (total scores can range between 7 and 35). After its publication, it was widely validated and translated into several languages, such as French, Italian, Romanian [21,22,23], Portuguese [24], Japanese [25] and Spanish [15,26,27,28].

#### 2.3.3. Personal Circumstances Regarding the Pandemic and Health

Respondents were asked if they had suffered COVID-19 and its sequelae (I have not had the disease; I have had the disease, but I am still suffering after-effects; and I have had the disease, but I do not have any after-effects). They were also asked how worried they were about the pandemic (from 1, not at all, to 4, a lot) and whether their satisfaction with life had changed (less, the same or more). Perceived health status was also assessed with the question “Usually, you would say that your health status is…”, with answers ranging from 5 (very good) to 1 (very poor). Adherence to three preventive measures, the “three Ws” (wearing a mask, watching physical distance and washing hands) [29], was determined in the questionnaire through the question “what do you do to keep yourself safe from COVID?” It included three measures: “Use a mask”, “Wash my hands” and “Keep a physical distance”. Item scores ranged from 1 (never) to 4 (always).

#### 2.3.4. Depression

For depression, an adapted and abbreviated version of the Geriatric Depression Scale (GDS-5) was used [30]. The scale uses five dichotomous (yes/no) questions (Are you basically satisfied with your life? Are you often bored? Do you prefer to stay in residence or in your room instead of going out and doing new things? Do you feel useless the way you are now? Do you think your situation is hopeless?).

#### 2.3.5. Emotions and Coping

To assess coping strategies, two different scales were employed. The Brief Resilient Coping Scale (BRCS) [31,32] is a four-item Likert-type scale that ranges from 0 (this sentence never happens to me) to 5 (this statement happens to me very often), with a maximum score of 20 (the higher the score, the greater resilience). It captures tendencies to cope with stress adaptively. Additionally, the Positive and Negative Affect Schedule (PANAS-Balance) [33], a 10-item scale validated in multiple settings, was applied. It measures emotional wellbeing through a self-reported questionnaire, and items are rated on a five-point scale from 1 (not at all) to 5 (very).

#### 2.3.6. Quality of Life

Quality of life (QoL) was measured with the FUMAT-24 scale [34,35], a scale specifically designed to assess QoL in LTC settings, and which includes 24 items grouped in 8 categories: emotional well-being, interpersonal relationships, material well-being, personal development, physical well-being, self-determination, social inclusion and rights. Items are ranged in a four-point Likert-type scale (from 1, always or almost always, to 5, never or hardly ever) and the results generate a global QoL index and percentages for each category.

### 2.4. Statistical Analysis

After testing for the outcome variable (FCV-19S) using the Kolmogorov–Smirnov test and the visual inspection of graphics, non-parametric statistics were applied. Descriptive statistics were used for the participants’ characteristics. The psychometric characteristics of the FCV-19S were explored using classical test theory (CTT) and Rasch analyses.

The CTT analysis included the calculation of the following psychometric attributes: data quality and acceptability, reliability (internal consistency) and construct validity (hypotheses testing).

For data quality and acceptability, the percentage of missing data (standard criterion: ≤15%) for the FCV-19S items, the mean, median, standard deviation (SD), skewness (criterion: −1 to +1), floor and ceiling effects of the FCV-19S items and the total score were calculated (criterion: ≤15%) [36].

For internal consistency, the inter-correlation among the FCV-19S items was calculated using Spearman’s rank correlation coefficients and an item homogeneity index. Internal consistency was explored by computing the Cronbach’s alpha (criterion: ≥0.70) [37].

Hypotheses testing comprised convergent and discriminative validity. Convergent validity was calculated using the Spearman’s rank correlation coefficients of the FCV-19S with the rest of the applied measures. Based on previous studies, a moderate correlation (rS = 0.30−0.50) [38] was hypothesized between the FCV-19S and emotional wellbeing (PANAS-Balance) [39,40], resilience (BRCS) [41], depression (GDS-5) [23] and quality of life (FUMAT-24) [42].

Discriminative or known-groups validity was explored by calculating the differences in the FCV-19S scores in the sample grouped by variables of interest: sex, age group, COVID-19 infection, education, depression, concern, satisfaction, perceived health status, face mask use, hand hygiene and physical distancing [43,44]. Mann–Whitney or Kruskal–Wallis tests were used to ascertain differences between groups.

Additionally, for the Rasch analysis, the following attributes were calculated: fit to the Rasch model, unidimensionality, item local independency, reliability (person separation index, PSI), threshold ordering, item–person distribution and differential item functioning (DIF) by age, sex, COVID-19 infection and COVID-19 diagnosis with or without sequelae. Modifications were performed iteratively until model fit and the other assumptions were achieved [45].

Fit to the Rasch model was considered achieved when there was a non-significant chi-square value with Bonferroni correction by number of items (*p* > 0.007), and when item and person fit residuals followed a distribution with a mean of 0 and a standard deviation of 1. In addition, fit residuals were expected to fall within the interval of −2.5 to +2.5.

For unidimensionality, a principal component analysis of residuals was calculated, and the person estimates of two sets of items defined in a principal component analysis of the residuals were compared through *t*-tests. For a scale to be unidimensional, the lower bound of the binomial confidence interval should overlap 5% [46,47].

An inter-item residual correlation lower than 0.30 of the mean correlations was used to ascertain item local independency. PSI measures reliability, and its interpretation is similar to that of Cronbach’s alpha [48]. Moreover, DIF occurs when, for the same construct level, two or more sample groups answer in a statistically different way. For each item, DIF was measured with a two-way analysis of variance (ANOVA) by different groups [49].

Category probability curves were used to identify ordered or disordered thresholds. Threshold ordering means that the participants use the response categories in an expected way, consistent with the construct continuum [50]. Furthermore, the item–person threshold distribution was visually inspected.

Statistical significance was set at a two-tailed *p*-value less than 0.05. CTT calculations were performed with the IBM SPSS Version 28.0 statistic software package. Rasch analyses were performed using the RUMM 2030 Rasch software.

## 3. Results

### 3.1. Participants’ Sociodemographic Characteristics

The mean age of the participants was 83.8 (standard deviation, SD: 8.9) years (Table 1). Most of the sample (63.1%) was female and 50.8% was widowed. The majority of participants (64.2%) had either primary education or no formal education. The sample had a mean of 6.06 (SD: 2.3) previous pathologies and was prescribed an average of 3.44 (SD: 1.4) medications.

Among the participants, 54.8% did not experience COVID-19 infection, 60.3% declared being quite or very worried about the pandemic and the majority (58.6%) perceived their health status as good or very good. For most of them (96.8%), life satisfaction remained the same or had decreased since the pandemic.

Almost all participants (n = 405, 90.6%) declared that they used facemasks, 425 (95.1%) washed their hands and 255 (57.5%) reported maintining physical distancing often or always.

### 3.2. Psychometric Properties According to CTT

The data quality and acceptability of the FCV-19S is shown in Table 2. The mean total FCV-19S score was 18.36 (SD: 8.28, range 7–35). The skewness of the total FCV-19S score was 0.207, and all item scores covered the full score range (1 to 5 points). Floor (14.5%) and ceiling (3.6%) effects were absent for the total score. Item 6, “I cannot sleep because I’m worrying about getting coronavirus-19”, presented the highest floor effect (42.1%) and item 1, “I am most afraid of coronavirus-19”, showed the highest ceiling effect (20.4%).

The Cronbach’s alpha was 0.94. The corrected item–total correlation ranged from 0.69 to 0.86. Additionally, the inter-item correlation was 0.46–0.86 for the total scale and the item homogeneity was 0.70.

Regarding convergent validity (Table 3), the correlation coefficient between FCV-19S total score and PANAS: Balance was rS = −0.25 (*p* < 0.01), and with GDS-5 it was rS = 0.17 (*p* < 0.01).

Regarding the known-groups validity (Table 4), significantly higher FCV-19S scores were found for women (*p* = 0.003), participants with the lowest educational level (*p* < 0.001), those with no previous COVID-19 infection (*p* = 0.001), those with a worse perceived health status (*p* < 0.001) and those with lower satisfaction with life after the pandemic (*p* < 0.001). There were also higher FCV-19S scores among participants with a higher use of facemasks (*p* < 0.001), hand hygiene (*p* = 002) and physical distancing (*p* = 0.003). There were no significant differences by age group, marital status or number of chronic diseases.

### 3.3. Psychometric Properties According to the Rasch Model

The Rasch analysis indicated that all items displayed ordered thresholds, with item local independency but no general fit to the Rasch model. After eliminating items 1 and 7 due to item misfit, a good fit to the Rasch model was observed, with χ^2^ (20) = 30.24, *p* = 0.019, PSI = 0.87 (Table 5 and Table 6).

Item 5 (“When watching news and stories about coronavirus-19 on social media, I become nervous or anxious”) showed DIF by sex, with women reporting higher levels of fear of COVID-19 (Appendix A). No DIF was observed by age group, COVID-19 infection or diagnosis (with or without sequelae).

The person–item threshold distribution showed a moderate floor effect and item threshold locations ranging from −3 to 3 logits. The distribution of person locations was close to normality (Appendix A, top part). Item thresholds (Appendix A, bottom part) represented most of the person distribution, although persons reporting lower and higher COVID-19 fear are less present.

## 4. Discussion

We present the measurement properties of the Spanish version of the FCV-19S scale in a population of older people living in LTC settings, as part of a study on the impact of COVID-19 in these environments. The use of two complementary methodological approaches, the CTT and Rasch model, provides a robust testing of the psychometric properties of the scale. In addition, the Rasch analysis enabled the conversion of raw scores into an interval scale, making possible to calculate how scores change and to use parametric statistics.

The scale had been validated for older adults, specifically [51]. It has been used before in LTC settings to measure fear of COVID-19 [18], but, to our knowledge, this is the first time the measurement properties of this tool have been evaluated in this context.

In our study, most of the patients were women, as in previous work [22]. The mean total score of the FCV-19S (18.36) shows that older people living in long-term facilities have a moderate level of fear. These data are in accordance with those of other national studies that have reported similar scores [17]. In addition, similar results have been found in older people in related environments [52]. However, our results varied from those of a national study that showed very high fear levels. This could be due to the period in which that study took place (the first pandemic wave) and the lack of vaccines at the time [23].

The FCV-19S meets the acceptability and data quality criteria we applied. The Cronbach’s alpha for the total score of the scale was very high, allowing for individual comparisons. This is in accordance with other studies in both Spanish [26,27,28] and other languages [24,53,54]. The Rasch analysis also showed high reliability.

The scale achieves model fit when items 1 (I am most afraid of coronavirus-19) and 7 (My heart races or palpitates when I think about getting coronavirus-19) are removed. Item 1 was found to be redundant, and item 7 was found to measure a different construct. The elimination of item 7 was also considered in another study [55]. However, they could be maintained in the scale, as they might provide useful clinical information, even if they do not contribute to the calculation of the total scale score.

Furthermore, items 3 (My hands become clammy when I think about coronavirus-19) and 6 (I cannot sleep because I am worrying about getting coronavirus-19) showed floor effects (having a lower percentage of respondents), as was the case in a study in the general population in Portugal [24]. This would indicate that only participants with a high level of fear would show physical signs (e.g., clammy hands or insomnia).

The convergent validity of the FCV-19S in LTC settings showed a significant correlation with depression (GDS-5) and an inverse correlation with emotional wellbeing (PANAS-Balance). The relationship between fear and depression has been hypothesized in other validation studies [21,22,56]. More resilience and greater satisfaction with life have also been associated with less fear [18,57,58], although in our study, the inverse correlation with the BRCS was not statistically significant.

Regarding discriminative validity, our results showed a significant association between fear, the following of preventive measures [43] and having a lower education [59]. These relationships are in line with results from other studies, and broadly support our validity hypotheses. Women had higher fear scores, as shown in previous investigations [25,60]. It has been reported that women have generally shown greater levels of stress and perceived personal risk during the pandemic, and thus higher fear levels [61,62].

The unidimensionality of the scale shown in the Rasch analysis supported its internal validity and is in concordance with other studies [63,64,65]. Nevertheless, the literature also reports that scale has a two-factor structure [66]. In general, items 3, 6 and 7 are considered somatic responses to fear, and items 1, 2, 4 and 5 are considered general or emotional responses.

Item 5 (When watching news and stories about COVID-19 on social media, I become nervous or anxious) showed DIF by sex, which is not consistent with the results of DIF analyses in other populations [67]. Moreover, a systematic review concluded that gender was not a significant factor that could affect the way individuals interpret any item in the FCV-19S [68]. Additionally, another study observed that most items displayed DIF across different countries and age groups, but not across gender [69]. However, a national study found that gender modulates fear of COVID-19 [15]. More research is needed to confirm these results in residents in LTC settings. In the meantime, differences by sex for this item should be regarded with caution.

### 4.1. Limitations and Strengths

Amongst the limitations of this study is the use of a cross-sectional design that provides information about a specific point in time. This approach, in the context of a changing pandemic, does not take into account the different measures and recommendations that were implemented depending on its evolution. Other limitations result from the exclusion criteria, e.g., it being impossible to know fear levels in residents with cognitive impairment. The study was also restricted to a regional level.

The study’s strengths include the use of two different methods to examine the psychometric properties of the FCV-19S, i.e., CTT and Rasch analyses. It also considered a representative sample of older people living in LTC facilities, a population that is not frequently taken into account and which is difficult to reach.

### 4.2. Implications

Research on the factors associated with fear in older people and other populations in different phases of the pandemic, measured with the FCV-19S, would be useful to design public health interventions adapted to the epidemiological situation and to prevent and ease fear. It could also help identify older people at risk of suffering from fear of COVID-19, and thus from negative consequences such as higher levels of stress and anxiety. The identification of risk factors and their impact on the FCV-19S will be part of a future investigation. Health education and communication strategies directed at these risk groups could lower fear levels without decreasing adherence to preventive measures.

The Age-Friendly Cities and Communities (AFCC) paradigm does not scrutinize fear of COVID-19 or other pandemics [70,71]. However, the perception of fear might have an impact on long-term care settings and limit participation in recreational activities, social connectedness or indoor and outdoor mobility.

## 5. Conclusions

The present paper presents the measurement properties of the Spanish version of the FCV-19S in a population of residents in LTC settings. This is a brief and unidimensional scale with satisfactory reliability and validity, which supports its use in effectively measuring fear in our specific population. In addition, our results suggest that women, people with a lower education level and a higher adherence to preventive measures might present more fear of COVID-19, although multivariate analyses would be necessary to confirm these results. Studies on the determinants of fear of COVID-19 could help to identify vulnerable groups of the population. One weakness is that one item presented DIF by sex; therefore, further research to test DIF by gender would be beneficial.

## Figures and Tables

**Table 1 ijerph-19-16183-t001:** Sample characteristics (n = 447).

Variables		n (%)	Min	Max	Mean (SD)
Sociodemographic Features					
**Age**			61	99	83.8 (8.9)
**Sex**	Female	282 (63.1)			
	Male	165 (36.9)			
**Marital status**	Married, living as a couple	64 (14.3)			
	Separated, divorced	46 (10.3)			
	Single	110 (24.6)			
	Widow/er	227 (50.8)			
**Level of education**	Cannot read or write	27 (6.0)			
	Can read or write, but no primary education	152 (34.0)			
	Primary education	108 (24.2)			
	Secondary education, vocational education	120 (26.9)			
	University education	40 (8.9)			
**Personal circumstances regarding the pandemic**					
**Coronavirus status**	I have not had the disease	233 (54.8)			
	I have had the disease, but I am still suffering after-effects	44 (10.4)			
	I have had the disease, but I do not have any after-effects	148 (34.8)			
**Worried about the COVID-19 pandemic**	Not at all	74 (16.7)			
	Somewhat	102 (23.0)			
	Quite a lot	97 (21.9)			
	A lot	170 (38.4)			
**Satisfaction with life before and during the pandemic (comparative perspective)**	Less	191 (43.1)			
	The same	238 (53.7)			
	More	14 (3.2)			
**Use of facemasks**	Never/Sometimes	42 (9.3)			
	Often/Always	405 (90.6)			
**Hand hygiene**	Never/Sometimes	22 (4.3)			
	Often/Always	425 (95.1)			
**Physical distancing**	Never/Sometimes	190 (42.5)			
	Often/Always	255 (57.5)			
**Health**					
**Self-assessment of health status**	Very poor/Poor	61 (13.7)			
	Fair	124 (27.7)			
	Good/Very good	262 (58.6)			
**Resident’s previous pathologies (num)**			1	12	6.1 (2.3)
**Intake of medications (num)**			1	8	3.4 (1.4)
**GDS**			0	5	1.7 (1.1)
**Quality of Life**					
**FUMAT-24 QoL**	Emotional well-being		1	3	9.6 (2.6)
	Interpersonal relationships		1	3	11.3 (1.3)
	Material well-being		1	3	10.7 (1.5)
	Personal development		1	3	10.4 (2.2)
	Physical well-being		1	3	9.1 (2.4)
	Self-determination		1	3	7.5 (2.0)
	Social inclusion		1	3	9.9 (2.0)
	Rights		1	3	11.1 (1.7)
	FUMAT Global Scores		45	95	79.6 (8.9)
**Sentiments and coping**					
**PANAS**	Positive affects subscale		5	20	11.4 (3.3)
	Negative affects subscale		5	20	8.7 (3.3)
	Balance (aggregation)		−12	15	2.7 (5.1)
**BRCS**	4–20		4	20	15.9 (4.1)

GSD: Geriatric Depression Scale; FUMAT-24 QoL: FUMAT quality of life adapted version of 24 items; PANAS: Positive Affect and Negative Affect Scale; BRCS: Brief Resilient Coping Scale.

**Table 2 ijerph-19-16183-t002:** Data quality, acceptability and item–total corrected correlation of the FCV-19S (n = 447).

FCV-19S Items	Mean	SD	Skewness	Observed Range	Floor Effect (%)	Ceiling Effect (%)	ITCC
1. I am most afraid of coronavirus-19.	3.13	1.43	−0.25	1–5	21.0	20.4	0.69
2. It makes me uncomfortable to think about coronavirus-19.	2.85	1.38	0.02	1–5	23.7	12.5	0.84
3. My hands become clammy when I think about coronavirus-19.	2.26	1.31	0.66	1–5	40.5	6.9	0.84
4. I am afraid of losing my life because of coronavirus-19.	2.77	1.46	0.08	1–5	30.6	14.0	0.81
5. When watching news and stories about coronavirus-19 on social media, I become nervous or anxious.	2.77	1.41	0.09	1–5	27.3	12.8	0.83
6. I cannot sleep because I am worrying about getting coronavirus-19.	2.23	1.31	0.68	1–5	42.1	6.9	0.79
7. My heart races or palpitates when I think about getting coronavirus-19.	2.35	1.33	0.48	1–5	38.7	6.3	0.86
**Total**	18.36	8.28	0.21	7–35	14.5	3.6	

ITCC: item total corrected correlation.

**Table 3 ijerph-19-16183-t003:** Convergent validity of the FCV-19S with related measures.

	Item 1	Item 2	Item 3	Item 4	Item 5	Item 6	Item 7	FCV-19S Total
**PANAS**: Positive affects	−0.07	−0.10 *	−0.21 **	−0.12 **	−0.11 *	−0.21 **	−0.21 **	−0.16 **
**PANAS**: Negative affects	0.34 **	0.24 **	0.07	0.27 **	0.24 **	0.08	0.10 *	0.23 **
**PANAS**: Balance	−0.27 **	−0.21 **	−0.19 **	−0.25 **	−0.22 **	−0.186 **	−0.204 **	−0.25 **
**BRCS**	−0.03	−0.01	0.08	0.04	0.02	0.05	0.05	0.04
**GDS-5**	0.11 *	0.14 **	0.20 **	0.14 **	0.14 **	0.18 **	0.16 **	0.17 **
**FUMAT**: Emotional well-being	−0.07	−0.07	−0.13 **	−0.12 *	−0.09	−0.14 **	−0.11 *	−0.11 *
**FUMAT**: Interpersonal relationships	0.03	0.10 *	0.12 *	0.03	0.09	0.12 *	0.15 **	0.11 *
**FUMAT**: Material well-being	−0.05	−0.04	−0.19 **	−0.08	−0.04	−0.17 **	−0.12 **	−0.10 *
**FUMAT**: Personal development	0.01	0.06	0.02	−0.04	0.02	−0.02	−0.02	0.02
**FUMAT**: Physical well-being	−0.15 **	−0.18 **	−0.26 **	−0.16 **	−0.14 **	−0.25 **	−0.27 **	−0.23 **
**FUMAT**: Self-determination	0.05	0.05	−0.01	−0.05	0.09	−0.02	−0.01	0.02
**FUMAT**: Social inclusion	0.04	0.05	0.04	0.03	0.09	0.04	0.07	0.06
**FUMAT**: Rights	−0.01	0.04	0.14 **	0.03	0.06	0.08	0.12 *	0.07
**FUMAT Total**	−0.06	−0.03	−0.12 *	−0.11 *	−0.02	−0.12 **	−0.09 *	−0.08

** *p* < 0.01; * *p* < 0.05.

**Table 4 ijerph-19-16183-t004:** Known-groups validity (only significant variables).

Variables		n	Mean	SD	Min	Max	*p*
**Sex**	Male	165	16.75	7.34	7	35	0.003 ^a^
	Female	282	19.30	8.66	7	35	
**Level of education**	Cannot read or write	27	22.74	8.39	7	35	<0.001 ^b^
	Can read or write, but no primary education	152	19.03	8.39	7	35	
	Primary education	108	19.28	8.39	7	35	
	Secondary education, vocational education	120	16.99	8.11	7	35	
	University education	40	14.48	5.59	7	28	
**Coronavirus status**	I have not had the disease	233	17.41	8.29	7	35	0.001 ^b^
	I have had the disease, but I am still suffering after-effects	44	21.91	7.11	7	31	
	I have had the disease, but I do not have any after-effects	148	18.47	8.25	7	35	
**Worried about the COVID-19 pandemic**	Not at all	74	11.93	5.87	7	28	<0.001 ^b^
	Somewhat	102	16.89	7.87	7	35	
	Quite a lot	97	19.02	7.15	7	31	
	A lot	170	21.85	8.14	7	35	
**Self-assessment of health status**	Very poor/Poor	61	18.60	8.37	7	35	<0.001 ^b^
	Fair	124	20.80	8.07	7	35	
	Good/Very good	262	17.15	8.12	7	35	
**Life satisfaction**	Decreased	191	21.19	8.33	7	35	<0.001 ^b^
	Same	238	16.31	7.64	7	35	
	Increased	14	16.07	7.44	7	30	
**Use of facemasks**	Never/Sometimes	42	13.64	7.15	7	29	<0.001 ^a^
	Often/Always	405	18.85	8.24	7	35	
**Hand hygiene**	Never/Sometimes	42	13.64	7.15	7	29	0.002 ^a^
	Often/Always	405	18.85	8.24	7	35	
**Physical distancing**	Never/Sometimes	190	17.01	8.34	7	35	0.003 ^a^
	Often/Always	255	19.40	8.09	7	35	
**Depression (GDS-5)**	Not depressed	30	19.60	8.19	7	35	0.003 ^b^
	Mild depression (1–2)	337	17.60	8.10	7	35	
	Severe depression (3–5)	80	21.10	8.52	7	35	

^a^ Mann–Whitney test, ^b^ Kruskal–Wallis test.

**Table 5 ijerph-19-16183-t005:** Individual item fit of the FCV-19S after removing items 1 and 7.

FCV-19S Item	Location	Standard Error	Fit Residual	χ^2^	*p* Value
2. It makes me uncomfortable to think about coronavirus-19.	−0.59	0.07	0.64	0.84	0.933
3. My hands become clammy when I think about coronavirus-19.	0.67	0.07	−1.74	11.47	0.022
4. I am afraid of losing my life because of coronavirus-19.	−0.35	0.06	0.98	4.99	0.289
5. When watching news and stories about coronavirus-19 on social media, I become nervous or anxious.	−0.38	0.07	−0.07	3.49	0.478
6. I cannot sleep because I am worrying about getting coronavirus-19.	0.66	0.07	−0.13	9.45	0.051

**Table 6 ijerph-19-16183-t006:** Goodness of fit to the Rasch model of the FCV-19S after removing items 1 and 7.

Attribute		Criteria	FCV-19S
**Item fit residual**	Mean	0	−0.06
	SD	1	1.05
**Person fit residual**	Mean	0	−0.48
	SD	1	1.16
**Item trait, χ^2^ (df)**		Low	30.24 (20)
**Interaction *p* value**		Non-significant (>0.007)	0.019
**Personal Separation Index**		>0.70	0.87
**Unidimensionality**	Independent *t*-tests	<5%	
	95% CI binomial	*	0.001–0.045

* Lower bound should be ≤ 0.05.

## Data Availability

Not applicable.

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
