# Peer review of "Validation of the Spanish Version of the Fear of COVID-19 Scale (FCV-19S) in Long-Term Care Settings"

_ijerph, 2022, doi:10.3390/ijerph192316183_

Round 1

Reviewer 1 Report

Title:

1) Remove the word properties

Abstract:

1) Describe the age of older people 

2) Clearly define lower education group.

Introduction:

1) What do you mean by current pandemic. Change this word. Because COVID 19 has different variants and waves all around the globe. I think in your study you are talking about COVID-19 in general.

2) Remove the word thanks to and replace it with "by"

3) Write something more about FCV-195 scale, who developed it first and when it has been developed and in which language? Also write how many languages it has been translated

Methods: 

1) Specifically write several languages, such as A, B, C. 

2) The measure section seems confusing to me. I did not understand that you used only FCV-195 scale or did you use other scales too? If you used multiple scales for assessing different outcomes, then I will suggest you add heading before discussing any scale. Moreover, present why you used each scale at the start. Our study used five different scales:

a. FCV-195 for assessing fear

b. GDS-5 for assessing depression

c. BRCS and PANAS for assessing coping strategies

d. FUMAT-24 for examining QoL.

3) Before writing CTT in second paragraph of statistical analysis, write its abbreviation in bracket in the first paragraph of sub-section 2.3

Discussion: 

1) Read and correct the sentence 2 and sentence 3 of 2nd paragraph of discussion: " but, although it has been used in LTC settings before [18], this is the 264 first time this tool is validated in this environment to our knowledge."

2) Did other studies also removed item 1 and item 7? If Yes acknowledge and it no then write why ?

3) Did other studies also observed ceiling effect with item 3 and 6? Discuss and cite 

4) Write strengths and limitations in separate heading

5) Add a heading which should explain what is the implication of this study?

Conclusion:

Do not add any citation in the conclusion section. It is more like implication.

Rework on conclusion.

Author Response

Please, find below the response to the comments. In the name of the rest of co-authors, we thank you for your review and appreciate your work.

Title:

1) Remove the word properties

Answer: We propose a new title, “Validation of the Spanish version of the Fear of COVID-19 Scale (FCV-19S) in long-term care settings”.

Abstract:

1) Describe the age of older people 

Answer: We have now described the age of the population in the abstract, and now it reads:

“Participants (n=447), aged 61-99 (mean 83.8; SD 89) years, were asked to complete the FCV-19S…”

2) Clearly define lower education group.

Answer: We have now defined in the abstract lower education group, and it reads:

“The mean FCV-19S score was 18.36 (SD 8.28, range 7-35), with higher scores for women, participants with lower education (primary or less) …”

Introduction:

1) What do you mean by current pandemic. Change this word. Because COVID 19 has different variants and waves all around the globe. I think in your study you are talking about COVID-19 in general.

Answer: The sentence has been modified to make a general reference to the pandemic, and now it reads:

“The pandemic caused by the SARS-CoV-2 virus has had severe repercussions…”

2) Remove the word thanks to and replace it with "by"

Answer: The sentence has been changed to “Fear of COVID-19 can be quantitatively measured by the Fear of COVID-19 Scale”.

3) Write something more about FCV-195 scale, who developed it first and when it has been developed and in which language? Also write how many languages it has been translated

Answer: The authors of the FCV-19S scale (Ahorsu et al.) have been specified in the text. The different languages the scale has been translated and validated have been added in the Methods section, as specified in the next comment. Now it reads:

“Fear of COVID-19 can be quantitatively measured by the Fear of COVID-19 Scale (FCV-19S) [14], a scale specifically developed by Ahorsu et al. in Farsi for the pandemic…”

“After its publication, it has been widely validated and translated into several languages, such as French, Italian, Romanian [21–23], Portuguese [24], Japanese [25], and Spanish [15, 26-28].”

Methods: 

1) Specifically write several languages, such as A, B, C. 

Answer: The different languages and their references have been added to the text, and now it reads:

“After its publication, it has been widely validated and translated into several languages, such as French, Italian, Romanian [21–23], Portuguese [24], Japanese [25], and Spanish [15, 26-28].”

2) The measure section seems confusing to me. I did not understand that you used only FCV-195 scale or did you use other scales too? If you used multiple scales for assessing different outcomes, then I will suggest you add heading before discussing any scale. Moreover, present why you used each scale at the start. Our study used five different scales:

  1. FCV-195 for assessing fear
  2. GDS-5 for assessing depression
  3. BRCS and PANAS for assessing coping strategies
  4. FUMAT-24 for examining QoL.

Answer: The following headings have been added to clarify the variables and scales used in the study:

2.3.1. Sociodemographic features, 2.3.2. FCV-19S, 2.3.3. Personal circumstances regarding the pandemic and health, 2.3.4. Depression, 2.3.5. Emotions and coping, 2.3.6. Quality of life.

3) Before writing CTT in second paragraph of statistical analysis, write its abbreviation in bracket in the first paragraph of sub-section 2.3

Answer: The abbreviation of the term has been added in the previous paragraph, and now it reads:

“The psychometric characteristics of FCV-19S were explored using Classical Test Theory (CTT)…”

Discussion: 

1) Read and correct the sentence 2 and sentence 3 of 2nd paragraph of discussion: " but, although it has been used in LTC settings before [18], this is the first time this tool is validated in this environment to our knowledge."

Answer: The sentence has been reformulated in order to make it clearer:

“The scale had been validated specifically for older adults. It has been used before in LTC settings to measure fear of COVID-19 [18], but, to our knowledge, this is the first time the measurement properties of FCV-19 are evaluated in this setting.”

2) Did other studies also removed item 1 and item 7? If Yes acknowledge and it no then write why ?

Answer: Another study considered eliminating item 7. It has been added to the discussion, and now it reads:

“… and item 7 was found to measure a different construct. The elimination of item 7 is also considered in another study [55]"

3) Did other studies also observed ceiling effect with item 3 and 6? Discuss and cite 

Answer: Although the evaluation of floor and ceiling effect has not been frequent in the previous literature, we have now cited a study that did found floor effect in items 3 and 6. The text was modified accordingly and now it reads:

“… items 3 (My hands become clammy when I think about coronavirus-19) and 6 (I cannot sleep because I am worrying about getting coronavirus-19) showed floor effects (having then a lower percentage of respondents), as in a study in general population in Portugal [24]”

4) Write strengths and limitations in separate heading

Answer: A new heading (4.1) about strengths and limitations has been added.

5) Add a heading which should explain what is the implication of this study?

Answer: We have added a section on the implications (4.2) of the study.

Do not add any citation in the conclusion section. It is more like implication.

Answer: We have deleted the references in this section.

Rework on conclusion.

Answer: We have expanded the conclusion paragraph, and now it reads:

“The present paper presents the measurement properties of the Spanish version of the FCV-19S in a population of residents in LTC settings. This is a brief and unidimensional scale with satisfactory reliability and validity, which supports its use to effectively measure fear in our specific population. In addition, our results suggest that women, people with a lower education level and higher adherence to preventive measures might present more fear of COVID-19, although multivariate analyses would be necessary to confirm these results. Studies on the determinants of fear of COVID-19 could help to identify vulnerable groups of population. One weakness is that one item presented DIF by sex; therefore, further research to test DIF by gender would be beneficial.”

Reviewer 2 Report

This interesting paper reported by Pilar Cárdenas Soriano et al. assesses the adequate reliability and validity of the CV-19S and supports its use to effectively measure fear in older people living in LTC. Indeed, this tool could be useful in identifying risk groups that require specific health education and effective communication strategies to reduce fear levels. However, while this study is useful, a number of concerns were considered in accepting this manuscript. I would be happy to accept this manuscript if the authors could sincerely address my questions and concerns within the framework of a major revision.

L2-3, the title is a little unclear about what you want to present in this study, please reconsider.

L7-23, I don't think the formatting is in line with the submission rules, please correct it. I think the part about keeping up appearances is also important, as the paper will be evaluated including the respective email addresses and corresponding authors.

L24, please define the term COVID-19; it is common to use coronavirus disease 2019.

L29-30, L37-38, I think these blanks should be removed in the abstract.

 L95, we suggest creating and adding a flowchart to facilitate reader comprehension. It would be more valuable to present the extent of the population covered and what the exclusion criteria and target groups are in an easy-to-understand manner.

 L98, where is the validity of the excluded 60-year-old criterion? We would like you to justify whether this is an internationally valid numerical criterion.

 L98, I would like you to justify why you were excluded due to cognitive impairment. Also, what specific symptoms of cognitive impairment are included in this study?

 L98, and this is important, I would like to see a clear statement regarding informed consent. It is not clear at which time series point in time and in which target population informed consent was obtained. Also, if informed consent could not be obtained from the patient, how was this handled (is there an assumption that, for example, a parent or guardian could write on behalf of the patient?)

 L101, the authors would like to see more specific information on sex, age, marital status, and level of education. For example, this means that the indicators should be separated into specific indicators such as sex (male or female) and listed in a clear manner.

 L143, please explain why you found that the distribution was not normal, either with data or statistical techniques.

 L185, define ANOVA, which I think is analysis of variance. Also, is this one-way or two-way? This should also be clearly stated.

 L190, is the significance level a two-tailed or one-tailed test?

 L142-192, I feel that the statistical analysis is well-designed and well described. However, my key concern is why the author only uses the descriptive statistical level summary to evaluate the general conclusions and does not carry out multivariate analysis (not considering the possibility of confounding variables). Although the statistical model needs to be flexible depending on the data fit, wouldn't it be more useful for the reader to observe the impact of each variable on the FCV-19S to further enhance the value of this study?

 Fig 1 and Fig 2, the writing and presentation of these figures is not common and very difficult to read, please correct this. I also think these figures would be more appropriate in the Supplementary Material.

 Discussions and Conclusions, well written. Thank you for your efforts. However, my concern is that L257-262 and L325-340 seem to be overstated and I would like you to weaken your arguments considerably. I don't think you can make so many claims with only descriptive evaluations of characteristics only. Neither can a statistical evaluation be said to be running a sophisticated analysis.

 L355-356, usually the approval number of the study period should be mentioned. Has the ethics committee been properly passed?

Author Response

Please, find below the response to the comments. In the name of the rest of co-authors, we thank you for your review and appreciate your work.

L2-3, the title is a little unclear about what you want to present in this study, please reconsider.

Answer: We propose a new title: “Validation of the Spanish version of the Fear of COVID-19 Scale (FCV-19S) in long-term care settings”.

L7-23, I don't think the formatting is in line with the submission rules, please correct it. I think the part about keeping up appearances is also important, as the paper will be evaluated including the respective email addresses and corresponding authors.

Answer: we have changed the formatting and now it follows the submission rules.

L24, please define the term COVID-19; it is common to use coronavirus disease 2019.

Answer: the definition of the term COVID-19 has been added, and now it reads:

“Fear of coronavirus disease 2019 (COVID-19) is one of the main psychological impacts…”

L29-30, L37-38, I think these blanks should be removed in the abstract.

Answer: The blanks have been removed.

L95, we suggest creating and adding a flowchart to facilitate reader comprehension. It would be more valuable to present the extent of the population covered and what the exclusion criteria and target groups are in an easy-to-understand manner

Answer: We do not know the number of long-term care settings or participants that have been excluded from the study. Nursing homes were selected in terms of probability proportional to their size (number of places), while participants were selected by simple random sampling (approximately ten persons in each nursing home). The nursing homes that decided not to participate were replaced by reserve nursing homes until the final number was reached.

The sample size was calculated with a maximum sampling error of ± 5%, for estimating percentages under the hypothesis of maximum variability (p=q=0.5) and at a confidence level of 95%.

All the information about the sample selection can be found in a previous study acknowledged in the Materials and Methods section, published in the same issue of this journal, and now it reads:

“Further information about the project development, descriptive data and sample selection criteria can be found in the publication by Rodriguez-Rodriguez et al. [20].”

L98, where is the validity of the excluded 60-year-old criterion? We would like you to justify whether this is an internationally valid numerical criterion.

Answer: We were interested in assessing fear of COVID-19 in older people living in long-term facilities. We established the criterion of 60 years of older to allow for recruiting also the younger groups of residents living in the included nursing homes.

 L98, I would like you to justify why you were excluded due to cognitive impairment. Also, what specific symptoms of cognitive impairment are included in this study?

Answer: The use of self-reported scales implies having cognitive abilities to understand and respond the questions, thus, we had to exclude those patients with a previous diagnosis of cognitive impairment. It can be now read on the participants heading:

“Participants were excluded if they were aged less than 60 years, suffered from cognitive impairment (since the use of self-reported scales implies having cognitive abilities to understand and respond the questions), or refused to sign an informed consent form.”

L98, and this is important, I would like to see a clear statement regarding informed consent. It is not clear at which time series point in time and in which target population informed consent was obtained. Also, if informed consent could not be obtained from the patient, how was this handled (is there an assumption that, for example, a parent or guardian could write on behalf of the patient?)

Answer:  The information on ethical approval and informed consent is presented at the end of the manuscript, as requested by the rules of the journal:

“Institutional Review Board Statement: The project fieldwork was approved in June 2020 by the Bioethics Committee of the Spanish National Research Council (CSIC), ref. 114/2020.

Informed Consent Statement: Informed consent was approved by the CSIC Bioethics Committee and obtained from all subjects involved in the study.”

All participants signed the informed consent by themselves, as they were cognitively intact.

L101, the authors would like to see more specific information on sex, age, marital status, and level of education. For example, this means that the indicators should be separated into specific indicators such as sex (male or female) and listed in a clear manner.

Answer: Different headings have been added to clarify the variables and scales used in the study. The categories of some of the variables (sex, level of education and marital status) have been specified under the correspondent heading.

L143, please explain why you found that the distribution was not normal, either with data or statistical techniques.

Answer: Non-parametric statistics were used due to the lack of normality of the FCV-19S variable. This was tested using the Kolmogorov-Smirnov test and visual inspection of graphics. The text now reads:

“After testing for normality the outcome variable (FCV-19S) using the Kolmogorov-Smirnov test and visual inspection of graphics, non-parametric statistics were applied”

 L185, define ANOVA, which I think is analysis of variance. Also, is this one-way or two-way? This should also be clearly stated.

Answer: The definition of the two-way ANOVA is now clearly stated in the text:

“For each item, DIF was measured with a two-way analysis of variance (ANOVA) by different groups [49].”

 L190, is the significance level a two-tailed or one-tailed test?

Answer: It is now stated that statistical significance was set at a two-tailed p-value less than 0.05:

“Statistical significance was set at a two-tailed p-value less than 0.05.”

L142-192, I feel that the statistical analysis is well-designed and well described. However, my key concern is why the author only uses the descriptive statistical level summary to evaluate the general conclusions and does not carry out multivariate analysis (not considering the possibility of confounding variables). Although the statistical model needs to be flexible depending on the data fit, wouldn't it be more useful for the reader to observe the impact of each variable on the FCV-19S to further enhance the value of this study?

Answer: A multivariate analysis focusing on the impact of each variable on the FCV-19S is out of the scope of this study, which is focused in the psychometric properties of the FCV-19S. The following information has been added to Implications and Conclusions:

“Research on the associated factors to fear in older people and other populations in different phases of the pandemic, measured through the FCV-19S, would be useful to design public health interventions adapted to the epidemiological situation to prevent and ease fear. It can also identify older people at risk of suffering from fear to COVID-19, and thus, negative consequences such as higher levels of stress and anxiety. Identification of risk factors and their impact on the FCV-19S will be part of a future investigation.”

“In addition, our results suggest that women, people with a lower education level and higher adherence to preventive measures might present more fear of COVID-19, although multivariate analyses would be necessary to confirm these results. Studies on the determinants of fear of COVID-19 could help to identify vulnerable groups of population.”

Fig 1 and Fig 2, the writing and presentation of these figures is not common and very difficult to read, please correct this. I also think these figures would be more appropriate in the Supplementary Material.

Answer: The mentioned figures are now presented as supplemental material in the Appendix and an explanation is offered as a Figure note. These graphics are commonly presented in Rasch analysis studies.

Discussions and Conclusions, well written. Thank you for your efforts. However, my concern is that L257-262 and L325-340 seem to be overstated and I would like you to weaken your arguments considerably. I don't think you can make so many claims with only descriptive evaluations of characteristics only. Neither can a statistical evaluation be said to be running a sophisticated analysis.

Answer: We based our conclusions on the analysis of the psychometric properties of the scale, not on the descriptive statistics of the sample. Moreover, the use of two different methodologies for the validation of the scale is recommended and offers robust results. However, we agree that the differences between groups would be further assessed using multivariate analyses, as now expressed in the conclusions:

“In addition, our results suggest that women, people with a lower education level and higher adherence to preventive measures might present more fear of COVID-19, although multivariate analyses would be necessary to confirm these results. Studies on the determinants of fear of COVID-19 could help to identify vulnerable groups of population.”

L355-356, usually the approval number of the study period should be mentioned. Has the ethics committee been properly passed?

Answer: As commented previously, this information is presented at the end of the manuscript, as requested by the rules of the journal.

Round 2

Reviewer 2 Report

Let me thank you for your effort in addressing all my comments. I am fully satisfied with your revision.